# LC-MS and GC-MS Data Fusion Metabolomics Profiling Coupled with Multivariate Analysis for the Discrimination of Different Parts of Faustrime Fruit and Evaluation of Their Antioxidant Activity

**DOI:** 10.3390/antiox12030565

**Published:** 2023-02-24

**Authors:** Ciro Cannavacciuolo, Stefania Pagliari, Chiara Maria Giustra, Sonia Carabetta, Werther Guidi Nissim, Mariateresa Russo, Paola Branduardi, Massimo Labra, Luca Campone

**Affiliations:** 1Department of Biotechnology and Biosciences, University of Milano-Bicocca, Piazza Della Scienza 2, 20126 Milan, Italy; 2Department of Agriculture Science, Food Chemistry, Safety and Sensoromic Laboratory (FoCuSS Lab), University of Reggio Calabria, Via dell’Università, 25, 89124 Reggio Calabria, Italy; 3NBFC, National Biodiversity Future Center, 90133 Palermo, Italy

**Keywords:** antioxidants, *Citrus autralasica*, novel foods, polyphenols, foodomics, multivariate data analysis, principal component analysis

## Abstract

The comparative chemical composition of different part of Faustrime fruits (peels, pulp, albedo, and seeds) extracted with different solvents was determined by GC-MS and UHPLC-HRMS QTof. The obtained data were also combined for their in vitro antioxidant activity by multivariate analysis to define a complex fingerprint of the fruit. The principal component analysis model showed the significative occurrence of volatile organic compounds as α-bisabolol and α-trans-bergamotol in the pulp and albedo, hexanoic acid in the seeds, and several coumarins and phenolics in the peels. The higher radical scavenging activity of the pulp was related to the incidence of citric acid in partial least square regression.

## 1. Introduction

*Citrus* is a genus belonging to the Rutaceae family, which is widespread in tropical areas of the world. Several species are largely consumed as traditional food and studied for their content of phytochemicals, such as polyphenolics, coumarins, limonoids, and volatile organic compounds (VOCs), which confer bitterness and fresh aroma to the fruits. Growing attention in nouvelle cuisine is drawn to “finger limes” (*Microcitrus australasica*), also known as “citrus caviar” for the appealing praline pulp. The species originated from Queensland and New South Wales rainforests where seven different varieties were classified by the Australian cultivar registration authority [1]. The native fruits’ phenotype is different in the color of pulp and peels, but a unique composition in limonene/isomenthone/citronellal was assessed as chemical volatile markers for the genus [2]. The large request for novel foods characterized by innovative flavor, texture, and appearance, is responsible for the extended production of finger limes in other regions of the world. In particular, the temperate climate and sunny weather of the Mediterranean area shows ideal environment conditions for growing the Faustrime cultivar, a hybrid of *Fortunella* spp., *Microcitrus australasica,* and *Citrus aurantifolia* [3,4]. The ripe fruit, harvested from July to November, appears with a vivid green peel and white pulp. The caviar-like pulp is captivating in gourmet cuisine to garnish fish or meat or beverages, releasing a pungent aroma and citric taste with a hint of pepper. The peels are used as a fresh spice to flavor dishes, while other parts (seeds and albedo) are usually discharged despite the high price of the fruit. The distinctive aroma of the fruit could be determined by the unusual volatile constituents in *Citrus*, such as limonene, β-phellandrene, and γ-terpinene, as reported in the Faustrime cultivar from Sicily [5]. Moreover, the occurrence of citronellal is reported as a major compound along with limonene, β-phellandrene, and γ-terpinene in a cultivar from southern Italy [3].

Polyphenols as flavonoids usually described in *Citrus* are reported for their antioxidant activity [6], cancer prevention [7], and metabolic syndrome alleviation [8]. Among the more than 170 molecules identified in *Citrus* with biological activity, Faustrime differed in the occurrence of eriocitrin, neoeriocitrin, diosmisin, and neodiosmisin content [9] without conferring a qualitative relevance between peel and pulp. Limonoids are oxygenated terpenoid compounds with several pharmacological properties, including anticancer, antimicrobial, and antidiabetic activity. This interesting class of bioactive compounds typically occurs in the peel, albedo, or seeds, conferring bitterness to the fruits [10,11]. Among triterpenoids, only limonexic acid was previously reported in the Faustrime cultivar [9]. Coumarins and furanocoumarins represent a consistent family of benzopyrone compounds commonly investigated in *Citrus*. Despite the assumption that pure coumarin could be hepatotoxic, coumarins naturally occurring in foods are allowed according to the new European aroma law [12] and studied for their anticancer activity [13]. Faustrime fruits are considered a high-value foodstuff for the growing gastronomic demand and healthy potential. Despite the large diffusion in cultivation crop and consumption of the fruit, specific studies reporting the metabolomic description of VOCs and secondary metabolites (polyphenols, coumarins, terpenes) are still lacking.

In line with the increasing demand for health-promoting products, several tools are largely used to describe the occurrence of bioactive phytochemicals in foods. Foodomics is an innovative approach to investigate functional foods integrating complex data from chemical identification such as mass spectrometry profiling and biological response [14]. In the current study, a comprehensive quali-quantitative investigation of VOCs and non-volatile compounds in Faustrime fruit was provided by means of GC-MS and UHPLC-QTOF-MS/HRMS analyses. The characterized chloroform and hydroalcoholic extracts, obtained from different parts of the fruit: peel (PE), pulp (PU), albedo (AL), and seeds (SE), were tested for antioxidant activity by measuring DPPH^•^ and ABTS^•+^ radical scavenging capacity. Finally, imaging techniques were used for the comprehensive fingerprint of the fruit by multivariate data analysis such as principal component analysis (PCA) and partial least square regression (PLS) to observe the fruit’s metabolomics and the chemical correlation with antioxidant activity.

## 2. Materials and Methods

### 2.1. Chemicals and Reagents

Analytical-grade methanol and n-hexane for extractions were obtained from Sigma-Aldrich (Milan, Italy). Ultrapure water (18 MΩ) was prepared by a Milli-Q purification system (Millipore, Bedford, MA, USA). methanol (MeOH), water, and formic acid with LC-MS grade were supplied by Romil (Cambridge, UK). Reference standards (>98% HPLC grade) apigenin, luteolin, rutin, coumarin, eriocitrin, isorhamnetin, hesperetin, naringenin, 1,1-diphenyl-2-picrylhydrazyl (DPPH^•^), and 2,2-azinobis-(3-ethylbenzothiazoline-6-sulfonate) (ABTS^•+^) reagents were purchased from Sigma-Aldrich (Milan, Italy). Standard stock solutions (1 mg mL^−1^) of each compound were prepared in methanol and stored at 4 °C. Diluted solutions and standard mixtures were prepared in MeOH/H_2_O·2:8 (*v*/*v*).

### 2.2. Plant Material and Extracts Preparation

The Faustrime fresh fruits (1 kg) were harvested in the ripening stage in the Napoli province of Italy (40.7085; 14.5258) in October 2022. All parts of the fresh fruits were manually separated and extracted with *n*-hexane (solid–liquid ratio 1:20 *w*/*v*) at room temperature for 2 nights for the extraction of volatile organic compounds (VOCs), as reported by Guzowska et al., 2022 [15]. For the recovery of the non-volatile compounds, PE, PU, AL, and SE were freeze-dried, pulverized, and extracted with chloroform via maceration (solid–liquid ratio 1:20 *w*/*v*) for 3 nights, 3 times for each solvent [16]. Subsequently, a mixture of methanol/water 50% (*v*/*v*) was used for recovery of polar fraction using the same extraction conditions reported above. The *n*-hexane extract was dried under nitrogen flow while chloroform and hydroalcoholic extracts were dried under a rotatory vacuum evaporator and stored at +4 °C, protected from light until analysis.

### 2.3. Semi-Quantitative GC-MS Analysis

The volatile composition of *n*-hexane extracts was investigated by gas chromatography–mass spectrometry (GC-MS_QP2010 Ultra-Shimadzu, Milan-Italy). Volatile compounds were separated using a capillary column MEGA SE52 (5% Phenyl, 95% Methyl Polysiloxane 30 m × 0.25 mm i.d., 0.25 µm film thickness, MEGA s.r.l, Milan, Italy). The oven temperature was held at 40 °C for 3 min and then increased at 6 °C/min to 200 °C and then increased at 15 °C/min to 250 °C for 5 min. The total run time was 38 min. The carrier gas was helium (purity > 99.999%). The injection temperature was set to 270 °C; the injection mode was split with a split ratio of 50. The mass scan spectra were recorded from 35–500 amu with the electron ionization (EI) source set to 70 eV. The MS ion source and interface temperature were set to 280 °C, and a solvent cut time of 0.5 min was selected. GCMS SOLUTION software (Shimadzu, Milan, Italy) was used for the control of equipment and data acquisition. The relative abundance (%) was calculated by dividing the area of each individual peak by the total sum of all detected peak areas multiplied by 100. Compound identification was made by comparing mass spectra with NIST11 library data.

### 2.4. Qualitative and Semi-Quantitative Analysis by UPLC-ESI/HRMS-UV

Qualitative and quantitative analyses of chloroform extracts and hydroalcoholic extracts were performed by using a system of liquid chromatography coupled with electrospray ionization (ESI) and high-resolution mass spectrometry (UPLC-ESI/HRMS), a Waters ACQUITY UPLC system coupled with a Waters Xevo G2-XS QTof Mass Spectrometer (Waters Corp., Milford, MA, USA), operating in both negative and positive ionization modes. The chromatographic separation of chloroform extracts was performed on a Biphenyl column (100 mm × 2.1 mm, 2.6 µm; Phenomenex, Torrance, CA, USA) held at 30 °C by using a mobile phase consisting of 0.1% (*v*/*v*) formic acid in water as solvent A and 0.1% (*v*/*v*) formic acid in acetonitrile as solvent B, a flow rate of 0.4 mL min^−1^, and a linear gradient from 0 to 20.0 min held in a range of 30–80% of the organic phase (B); from 20.0 to 25.0 min rising to 90% B; from 25.0 to 30.0 min rising to 95% B; 3 min of column washing (95% B) and 5 min of column equilibration at 30% B was performed before the next sample injection. For the chromatographic separation of hydroalcoholic extracts, a linear gradient from 5–10% B in 2 min, rising to 80% B in 17 min, rising to 95% B in 18 min; after each run, 3 min of column washing (95% B) and 5 min of column equilibration (5% B) was used. The autosampler was set to inject 5 µL of each extract at a concentration of 0.5 mg mL^−1^. For the ESI source, the following experimental conditions were used: electrospray capillary voltage 2.5 kV, source temperature of 150 °C, and desolvation temperature of 500 °C. MS spectra were acquired by full range acquisition covering a mass range from 50 to 1200 *m*/*z*. The MS/HRMS analysis was provided by data-dependent scan (DDA). Experiments were performed by selecting the first and the second most intense ions from the HRMS scan event and submitting them to collision-induced dissociation (CID) by applying the following conditions: a minimum signal threshold at 250, an isolation width at 2.0, and normalized collision energy at 30%. Both in full and in MS/HRMS scan mode, a resolving power of 30,000 was used. Compound deconvolution was attributed via MS-DIAL 4.90 open-source software comparing MS/HRMS spectra with “ESI(-)-MS/HRMS from authentic standards (9033 unique compounds)” library data and confirmation with literature reports.

For semi-quantitative analysis, stock solutions (1 mg mL^−1^) of each used compound as external standards were prepared by dissolving each compound in a solution of methanol. Increasing solution concentrations (0.1, 0.5, 1, 5, 10, and 50 µg mL^−1^) were prepared for the calibration curve construction by UHPLC-UV, wavelength set at 280 nm. In particular, 5 µL of each standard solution at each concentration was injected in a technical triplicate. Calibration curves were obtained by using linear regression using the Excel 2016 Software (Microsoft Corporation, Washington, DC, USA), considering the area of external standards against the known concentration of each compound. The obtained calibration curves showed good linearity (with correlation coefficients (R^2^) ranging from 0.9969 to 0.9999, respectively, and all information is provided in Appendix A. The quantitative data are expressed as mg g^−1^ of dried extract (EXT). The MassLynx software version 4.2 (Waters, Wilmslow, UK) was used for instrument control, data acquisition, and processing.

### 2.5. Determination of DPPH and ABTS Radical Scavenging Activity

The free radical scavenging activities of extracts were determined using the 1,1-diphenyl-2-picrylhydrazyl radical (DPPH^•^) and 2,2-azinobis-(3-ethylbenzothiazoline-6-sulfonate radical cation) (ABTS^•+^) by the slightly modified methods previously described [17,18]. For the determination of DPPH^•^ scavenging activity, an aliquot (50 µL) of the extract (1 mg mL^−1^) or standard solution (2.5–10 µg mL^−1^) was added to 950 µL of prepared radical solution (0.14 mM). After darkness incubation for 30 min at room temperature, samples were read by spectrophotometer at a wavelength of 515 nm. For the ABTS^•+^ scavenging activity, the reaction was initiated by the addition of 50 µL of extract in 950 µL of diluted ABTS (14 mM) of each sample solution. The spectrophotometer was set with a wavelength of 734 nm. Determinations were repeated three times. The absorbance was calculated for each concentration relative to a blank absorbance (methanol) and was plotted as a function of the concentration of compound or standard, 6-hydroxy-2,5,7,8-tetramethylchroman-2-carboxylic acid (Trolox). The antioxidant activity was expressed as Trolox equivalents (TE) value representing the µmol of a standard Trolox solution exerting the same antioxidant capacity as 1 mg mL^−1^ of the tested extract.

### 2.6. Data Fusion Approach and Multivariate Data Analysis

A large dataset containing GC-MS and UPLC-MS data of all investigated compounds was organized for multivariate data analysis with a ‘Low-level’ data fusion approach [19]. The GC-MS matrix was made by 59 variables (metabolites tentatively identified in *n*-hexane extracts) and 4 observations (PE, PU, AL, and SE), while the LC-MS matrix contained 35 variables for the metabolites tentatively identified in chloroform and hydroalcoholic extracts of the same observations. To observe the comprehensive distribution of the volatile and polar compounds in four selected parts, the integrations of GC-MS and LC-MS signals were selected in the matrices and normalized by average function to balance the areas under the curve (AUC) obtained by the two tools. Finally, the total matrix was described by 94 variables (metabolites tentatively identified by GC-MS and LC-MS analysis) and 4 observations (PE, PU, AL, and SE). The obtained dataset was imported into SIMCA^®^ (Sartorius, Goettingen, Germany) 17 for statistical analysis and modeling of score and loading plots. PCA and PLS models were described by two main principal components, scaled by Pareto mode and Autofit function. For reproducibility of the analysis, three extraction replicates and three replicate injections per sample extract were provided. LC-MS and GC-MS experiments were randomized; several injections of extraction solvent (blank) and a mix of all the samples (QC) were periodically injected for each five-acquisition analysis. The injection of QC ensures the reproducibility and stability of the MS signals during the acquisition batch and also guarantees the robustness of the chemometric results.

### 2.7. Statistical Analysis

Each GC-MS and LC-MS experiment was performed in triplicate. The variance in assessing differences between groups was evaluated by one-way ANOVA (considered to be significant at *p* ≤ 0.05) on Microsoft Excel 2016. DPPH^•^ and ABTS^•+^ radical scavenging tests were carried out in triplicate and the results were expressed as mean ± standard deviation (SD). Multivariate data analysis was performed by principal component analysis (PCA) and partial least square (PLS) imaging techniques operating the linear regressions on the mean-scaled dataset to select the two main principal components. The choice of principal components was established based on the fitting (R2X, 0.623) and predictive (Q2X, 0.078) values. For PCA models, the chosen PC1 and PC2 explained 62.3% and 22.6% of the variance, respectively, for a total explained variance of 84.9%. The PLS analysis showed a distinct separation (R2Y, 0.702) and good predictability (Q2, 0.365). No outliers were detected based on Hotelling’s T2 and Q residual statistical test (confidence level of 95%).

## 3. Results and Discussion

### 3.1. Composition of Volatile Organic Compounds in Faustrime Fruits

The profiles of *n*-hexane extracts of PE, PU, AL, and SE are reported in the Appendix A and are described in detail in Table 1. Overall, 59 VOCs were identified and the relative abundance for each compound was reported. Monoterpenes were the most abundant VOCs identified, followed by sesquiterpenes. Generally, a citronellal/limonene/linalyl acetate chemotype can be considered for the investigated cultivar, highlighting differences in abundance ratios with previous studies [5,9]. The occurrence of linalyl acetate was reported for the first time in Faustrime. Significative differences in VOCs occurring in the four parts of the fruit were investigated in detail. The peel was the part of the fruit highlighting a larger number of GC-MS signals. In particular, citronellal was the most relative abundant monoterpene hydrocarbon detected in the peel (23.47%), followed by limonene, citronellol, and linalyl acetate (12.99%, 10.65%, and 7.65% respectively). An inverse citronellal/limonene ratio was registered for the peel of the investigated cultivar compared to the studies reported in the literature [5,8]. Differently, the pulp of the fruit highlighted the main content of linalyl acetate and limonene (18.23% and 11.40%, respectively), along with a higher content of *trans*-piperitol (1.83%) conferring the typical hint of paper to the culinary used pulp, compared to the peel, in contrast with Cioni et al., 2022 [9]. The detailed investigation of albedo and seeds of Faustrime was reported here for the first time. Linalyl acetate, citronellol, and limonene were the higher relative terpenes detected in the albedo (10.60%, 9.68%, and 9.03%, respectively) while cyclohexanol and hexanoic acid were mainly relative detected in the seeds (19.82% and 9.59%, respectively).

### 3.2. LC-HRMS Analysis of Faustrime Fruit

The chemical profiling of PE, PU, AL, and SE were investigated using UHPLC-(ESI)-DAD-MS/HRMS experiments. The chromatographic profiles of the chloroform and hydroalcoholic extracts of the four parts showed some differences, as reported in Appendix A.

The detailed identification of compounds was performed in several steps. Initial chemical deconvolution was performed by MS-DIAL software matching with the open access library provided by the website (http://prime.psc.riken.jp/compms/msdial/main.html, accessed on 11 December 2022); then, identities were carefully confirmed by comparing the analyzed retention times, HRMS mass spectra, fragmentation patterns, and molecular formulas with the literature (Table 2 and Table 3). Compounds 72–73, 80, 82, and 87 were confirmed based on the injection of reference standards. The following approach allowed the tentative identification of three coumarins and five furocoumarins reported for the first time in *Citrus australasica*; seven limonoids, despite only limonexic acid being previously reported in Faustrime [9]; and three organic acids. Moreover, six compounds related to the class of C-glycoside flavonoids, nine compounds ranked as *O*-glycoside flavonoids, and 2-hydroxymethylglutaryl flavonoids (HMG-flavones) were described, most of them reported for the first time in the specie.

#### 3.2.1. Identification of Coumarins and Furocoumarins

The coumarins detected in the chloroform extracts showed a good MS response in positive mode, and they generally exhibited a distinct fragment ion [M−CO_2_+H]^+^ from the fragmentation of pyran-2-one, which could be used as a diagnostic ion for the characterization of coumarins and furocoumarins [12] in the fruit, as in the case of the peaks 60–61 and 71 (Table 2). In the case of furocoumarins 62–66, fragment ions [M-CH_3_+H]^+^, [M-2CH_3_+H]^+^, and [M-2CH_3_-CO+H]^+^ occur in the MS/HRMS spectra, revealed by the neutral loss of 15 Da, 30 Da, or 58 Da, respectively (Table 2). To the best of our knowledge, the identified coumarins (compounds 60–66) were described for the first time in *Citrus australasica*.

#### 3.2.2. Identification of Limonoids

Limonoids are a class of oxygenated triterpenoids widely reported in *Citrus*. To distinguish them from other types of *Citrus* components in MS analysis, limonoids showed a distinct elemental composition (26–28 carbons) and degree of unsaturation (11–12), as well as a prominent ion at *m*/*z* 161.0595 in the positive mode MS^2^ spectra and *m*/*z* 159.0233 in negative mode MS^2^ spectra, which could be used as diagnostic ions for the class [20]. In the present study, compounds 67–70 were tentatively assigned to limonin, obacunoic acid, methyl nominilate, and veprisone, respectively, investigated in chloroform extracts (Table 2). Moreover, compounds 92–95 were tentatively identified in hydroalcoholic extracts as limonexic acid, citrusin, limonin and isobacunoic acid, respectively (Table 3). Limonin was detected in both chloroform and hydroalcoholic extracts and unambiguously identified by the injection of reference standards. The occurrence of limonexic acid was previously reported for Faustrime fruits by Cioni et al., 2022 [9], while the other investigated limonoids were reported for the first time in the specie.

**Table 2 antioxidants-12-00565-t002:** LC-MS data for coumarins, furocoumarins, and limonoids detected in chloroform peel, pulp, albedo, and seed extracts of Faustrime fruit.

n.	t_R_	[M+H]^+^	Compound	Class	ppm	Formula	MS/HRMS	Ref
60	6.11	163.0408	4-hydroxycoumarin	coumarin	8.0	C_9_H_6_O_3_	119.0075; 87.0074	[21]
61	6.96	193.0512	scopoletin	coumarin	5.7	C_10_H_8_O_4_	178.0267; 149.0314 133.0293; 105.0336	[22]
62	10.15	203.0344	bergaptol	furocoumarins	8.7	C_11_H_6_O_4_	175.0391; 147.0442 131.0491; 119.0492 103.0542	[22]
63	10.55	203.0364	xanthotoxol	furocoumarins	8.9	C_11_H_6_O_4_	175.0418; 147.0443 131.0497; 129.0337	[22]
64	13.39	187.0406	psoralen	furocoumarins	5.9	C_11_H_6_O_3_	131.0497; 115.0545 103.1522	[23]
65	15.00	217.0515	xanthotoxin	furocoumarins	6.5	C_12_H_8_O_4_	202.0258; 161.0612 146.0359; 131.0497 118.0431	[23]
66	16.92	247.0605	isopimpinellin	furocoumarins	1.4	C_13_H_10_O_5_	232.0360; 217.0154 189.0186; 161.0249	[24]
67	17.14	471.2038	limonin	limonoids	4.0	C_26_H_3_O_8_	425.1989; 367.1981 213.0987; 161.0612 133.0641	standard
68	17.64	473.5294	obacunoic acid	limonoids	0.8	C_26_H_32_O_8_	455.2148; 427.2138 369.2047; 341.2100 243.1015; 187.0770 161.0586	[20]
69	19.34	547.2559	methyl nomilinate	limonoids	2.9	C_29_H_38_O_10_	529.2419; 487.2338 469.2249; 369.2047 351.1956; 341.2100 187.0770; 175.0743 161.0612	[25]
70	20.22	487.2338	veprisone	limonoids	1.2	C_27_H_34_O_8_	369.2047; 215.1087 161.0586; 133.0647	[26]
71	21.44	163.0408	umbelliferone	coumarin	8.0	C_9_H_6_O_3_	135.0451; 133.0269 119.0143; 120.0209 107.0499; 105.0336	[27]

#### 3.2.3. Identification of HMG-Hexoside and HMG-Flavonoids

The first chromatographic region of hydroalcoholic extracts, reported in Appendix A, showed organic acids (compounds 72, 73 and 74). Citric acid and ascorbic acid were assigned to compounds 72 and 73, respectively, for the overlapping of retention time and HRMS spectra with standards. Compound 74 showed a deprotonated ion [M-H]^−^ at *m*/*z* 365.1440 and typical MS/HRMS signals of HMG-moiety as the product ion [M-144-H]^−^ at *m*/*z* 221.1025, along with the diagnostic residue at *m*/*z* 125.0255, confirming the identity of propyl HMG-hexoside [22]. The fragmentation patterns of compounds 89 and 91 regarded the structure of flavones with HMG moiety, confirmed by the characteristic neutral loss of 144 Da (C_6_H_8_O_4_) and a diagnostic ion at *m*/*z* 125.02272 (HMG–H_2_O) in the MS/HRMS spectra. In particular, compound 89 showed a [M-H]^−^ at *m*/*z* 651.1567 (C_29_H_32_O_17_). The fragments [M-144-H]^−^ at *m*/*z* 507.1143 and [M-162-144-H]^−^ at *m*/*z* 345.0618 were observed in MS/HRMS spectra suggesting the presence of 3-hydroxy-3-methylglutaryl moiety attached to the glycosyl group and the aglycone tetrahydroxy-dimethoxyflavone. A similar MS^2^ pattern was observed for compound 91, showing a HRMS deprotonated ion [M-H]^−^ at *m*/*z* 675.1843 (C_34_H_38_O_20_) never found in the MS database and the literature. In detail, HRMS and MS^2^ Spectra are shown in Appendix A. The fragments [M-144-H]^−^ at *m*/*z* 621.1473, and [M-(2x144)-H]^−^ at *m*/*z* 477.1016 highlight the occurrence of two HMG residues on the glycosylated isorhamnetin (C_22_H_22_O_12_) whose molecular formula was obtained by high-resolution MS^2^ patterns. The high-resolution fragment at *m*/*z* 315.0488 [M-(2x144)-162-H]^−^ confirms the occurrence of isorhamnetin as an aglycon scaffold. To the best of our knowledge, the tentatively identified di-HMG-hexoside-flavone is observed for time and tentatively attributed to isorhamnetin-di-HMG-*O*-glycoside.

#### 3.2.4. Identification of C-Glycoside Flavonoids

Compounds 75, 77, 78, 79, 81, and 86 are ranked in the classes of C-glycoside flavonoids for the typical MS^2^ patterns, highlighting diagnostic losses of M-90, M-120, and M-150 Da from the cross-ring cleavage of C-sugar of [M-H]^−^ ions (Table 3). The occurrence of C-glycoside flavonoids lucenin 2 (75), homoorientin (78), stellarin 2 (79), and scopanin (86) was reported for the first time in the species.

#### 3.2.5. Identification of *O*-Glycoside Flavonoids

Differently, the compounds 76, 80, 82–85, 87–88, and 90 were characterized as *O*-glycoside flavonoids by the more favorable loss of glycoside units as dehydrated forms in the product ion spectra (Table 3). Fragments of [M-162-H]^−^ and [M-146-H]^−^ are commonly found for the presence of hexose and pentose sugars, respectively, while di-glycosides can also be detected with [M-324-H]^−^ and [M-308-H]^−^ fragments in the MS^2^ pattern attributed to sucrose or rhamnose moieties, respectively. Rutin, neoeriocitrin, and naringenin 7-*O*-rutinoside were identified by the injection of standard, confirming the identification of neoeriocitrin for the first time in Faustrime.

**Table 3 antioxidants-12-00565-t003:** LC-MS data for organic acids, phenolic compounds, and limonoids detected in chloroform peel, pulp, albedo, and seed extracts of Faustrime fruit.

n.	tr	[M-H]^−^	Compound	Class	ppm	Formula	MS/HRMS	Ref
72	1.04	191.0187	citric acid	organic acid	0.26	C_6_H_8_O_7_	111,0075; 87.0074	standard
73	2.51	175.1324	ascorbic acid	organic acid	1.32	C_6_H_8_O_7_	115.0043; 87.0043	standard
74	7.72	365.1440	propyl HMG-hexoside	organic acid hexoside	−2.2	C_15_H_26_O_10_	221.1020; 161.0461 125.0255; 113.0421	[22]
75	8.38	609.1467	lucenin 2	C-di-glucosyl flavonoid	1.8	C_27_H_30_O_16_	519.1135; 489.1046 429.0822; 399.0700 369.0594	[28]
76	9.02	639.1581	isorhamnetin-3,7-di-*O*-glucoside	O-di-glucosyl flavonoid	1.3	C_28_H_32_O_17_	519.1521; 477.1061 357.0322; 315.0561	[29]
77	9.24	593.1518	vicenin 2	C-di-glucosyl flavonoid	2.1	C_27_H_30_O_15_	473.1072; 383.0711 353.0654	[29]
78	9.85	447.0903	homoorientin	C-glucosyl flavonoid	5.4	C_21_H_20_O_11_	399.0721; 357.0613 327.0524; 299.0533	[30]
79	10.57	623.1597	stellarin 2	C-di-glucosyl flavonoid	−2.4	C_28_H_32_O_16_	533.1282; 503.1197 413.0865; 383.0760	[31]
80	10.87	609.1467	rutin	O-glucosyl-rhamnosyl flavonoid	1.8	C_27_H_30_O_16_	301.0337	standard
81	11.07	431.0951	isovitexin	C-glucosyl flavonoid	3.5	C_21_H_20_O_10_	341.0644; 323.0550 311.0563; 283.0600	[32]
82	11.34	595.1683	neoeriocitrin	O-glucosyl-rhamnosyl flavonoid	3.4	C_27_H_32_O_15_	459.1112; 287.0564 151.0007	standard
83	11.40	593.1581	luteolin-*O*-rutinoside	O-glucosyl-rhamnosyl flavonoid	2.0	C_27_H_30_O_15_	285.0374	[9]
84	12.15	623.1597	isorhamnetin-3-*O*-rutinoside	O-glucosyl-rhamnosyl flavonoid	−2.4	C_28_H_32_O_16_	315.0488	[33]
85	12.30	477.1016	isorhamnetin-3-*O*-glucoside	O-glucosyl flavonoid	1.9	C_22_H_22_O_12_	357.0584; 314.0392 285.0490; 271.0228 243.0278; 151.0007	[33]
86	12.38	461.1097	scoparin	C-glucosyl flavonoid	2.2	C_22_H_22_O_11_	371.0762; 341.0644 298.0462	[34]
87	12.55	579.1719	naringenin-7-*O*-rutinoside	O-glucosyl-rhamnosyl flavonoid	0.9	C_27_H_32_O_14_	271.0599; 151.0032	standard
88	12.87	607.1673	diosmetin-7-*O*-rutinoside	O-glucosyl-rhamnosyl flavonoid	1.6	C_28_H_32_O_15_	299.0546; 284.0254	[29]
89	13.16	651.1567	tetrahydroxy-dimethoxyflavone HMG-hexoside	HMG-hexoside-flavonoid	3.2	C_29_H_32_O_17_	607.1673; 589.1590 549.1228; 507.1143 345.0618	[35]
90	13.58	609.1821	hesperetin-7-*O*-rutinside	O-glucosyl-rhamnosyl flavonoid	0.3	C_28_H_34_O_15_	301.0639; 286.0824	[29]
91	13.99	765.1843	isorhamnetin-di-HMG-*O*-glycoside	di-HMG-hexoside-flavonoid	−4.1	C_34_H_38_O_20_	621.1473; 477.1016 315.0488; 125.0233	-
92	14.88	501.1788	limonexic acid	limonoid	−3.8	C_26_H_30_O_10_	457.1872; 413.1948 59.0214	[22]
93	16.78	545.2000	citrusin	limonoid	−4.2	C_28_H_34_O_11_	501.2155; 457.2222 397.1988; 373.2013; 125.0233; 59.0233	[22]
94	17.70	469.2869	limonin	limonoid	1.5	C_26_H_30_O_8_	451.1738; 249.1307 229.1233; 59.0232	standard
95	17.78	471.2027	isoobacunoic acid	limonoid	1.7	C_26_H_32_O_8_	453.1898; 261.1464 231.0988; 59.0342	[22]

### 3.3. Semi-Quantitative Analysis of Coumarins and Flavonoids

Qualitative analysis was combined with a semi-quantitative approach to highlight the different distribution of secondary metabolites among the different parts of the matrix. Semi-quantitative analysis was performed using a UPLC-UV at a wavelength of 280 nm, and quantification was carried out by using calibration curves in a range of 0.1–50 µg mL^−1^ with 8 standards, reported in Appendix A, used for the classification of all major phenolic compounds and coumarins. Compounds for which authentic standards were not available were quantified into standard equivalents using the most chemically related standard available.

The results of the semi-quantitative analysis (Table 4) showed the different amount distribution of coumarins and phenolics in PE, PU, AL, and SE, relatively. In detail, Faustrime PE showed the highest phenolic compounds and coumarin concentration. Among these, the highest relative amounts are referred to compounds 60, 62, 64, 76, 80, 87 and the newly observed compound 91 (Table 4). In addition, the absence of coumarins was highlighted in the albedo portion compared to the peel and pulp, where they were found and quantified (compounds 60 to 66). However, neoeriocitrin prevailed in the albedo (10.92 mg/g EXT) compared to the other parts of the fruit, while diosmetin-7-*O*-rutinoside was mainly contained in the pulp (11.33 mg/g EXT). The UPLC-UV profile of the seeds did not show a considerable signal of non-volatile compounds for the quantification (Appendix A).

### 3.4. Antioxidant Properties of Bioactive Containing Extracts

The comparison between chloroform and hydroalcoholic extracts of the different parts of the fruit (PE, PU, AL, and SE) to determine the antioxidant activity was performed by DPPH^•^ and ABTS^•+^ assays. Generally, the hydroalcoholic extracts showed higher activity than chloroform extracts. In particular, the hydroalcoholic extracts of the pulp and albedo showed a higher capacity to neutralize both DPPH^•^ and ABTS^•+^ radicals (Table 5). The peel showed a lower scavenging capacity with TE values of 2.75 ± 0.05 and 8.22 ± 0.32 for DPPH^•^ and ABTS^•+^, respectively, whereas seeds showed very low antioxidant activities (Table 5).

### 3.5. Combined Multivariate Data Analysis by PCA and PLS

With the aim to visualize a complex set of chemical data and correlating the radical scavenging capacity to specific metabolites contained in the investigated extracts, a supervised multivariate data analysis was carried out. Specifically, a combined data analysis was performed by principal component analysis (PCA) and partial least squares–discriminant analysis (PLS-DA) methods to discriminate between the samples and model the complex metabolome of Faustrime.

Data fusion furnishes statistical results to obtain the maximum number of correctly classified samples by combining the data obtained from different instrumental techniques. One of the strategies is ‘low-level’ data fusion, where the two blocks of data (LC and GC matrices) obtained from the mass spectrometry techniques are pre-treated and mixed in a single dataset [36]. After a detailed targeted analysis of the GC-MS and LC-MS profiles of Faustrime’s VOCs and non-volatile compounds, in the second stage, the combination of the data from the different instruments by means of ‘low-level’ data fusion strategy was performed to build a comprehensive metabolomic fingerprint. The ‘low-level’ data fusion strategy consists of concatenating the individual LC-MS and GC-MS data matrices, opportunely pre-treated, and finally, concentrating the resulting data in a single block [19]. The different data matrices corresponding to the GC-MS profiles related to the *n*-hexane extracts of the four different parts of the fruit integrated with HPLC-MS profiles recorded for chloroform and hydroalcoholic extracts were concentrated using the mean-centering pre-treatment to normalize data obtained by different analytical tools [19].

The first principal component (PC1) divided the peel on the left part (PC1 < 0) of the PCA biplot and the group containing albedo, pulp, and seeds (PC1 > 0) on the right quadrant of the biplot (Figure 1A). In detail, the cluster of albedo and pulp was plotted on the upper right quadrant (PC1 > 0, PC2 > 0), discriminated for the content of several VOCs, such as α-bisabolol, 2-menthen-1-ol, and α-*trans*-bergamotol, while seeds laid on the opposite quadrant along the second principal component (PC1 > 0, PC2 < 0) for the incidence of hexanoic acid. The presence of all the identified non-volatile compounds and most of the VOCs resulted in the discrimination of the peel. The incidence of the metabolites on that cluster depended on their distance from the origin, highlighted in detail by the contribution bars of Figure 1B. Citronellal is the main constituent of volatile chemotype interfering in the peel discrimination, along with the detected HMG-flavonoids, lucenin 2, stellarin 2, and vicenin 2 C-glycoside flavonoids, scopoletin and psolaren (coumarin and furocoumarin, respectively) and caryophyllene, α-bergamotene (terpenes), mainly occurring in peels. PCA loading and PCA score plots are shown in Appendix A, respectively.

Partial least square regression (PLS) was performed for the attribution of classes to the chloroform and hydroalcoholic extracts based on the different capacities to neutralize DPPH^•^ and ABTS^•+^ radicals among the extracts of peel, pulp, and albedo. The seeds are not considered because of their low activity in the spectrophotometric tests. PLS showed the separation along the first principal component (PC1) between the peel (PC1 > 0) against pulp and albedo (PC1 < 0), while the second principal component (PC2) separated the pulp (PC2 < 0) and albedo (PC2 > 0) (Figure 2). The biplot in Figure 2A shows metabolites that are potentially significant in the discrimination of the parts based on contributions and reliability of the separation observed in the score scatter plot (Appendix A). The incidence of citric acid determined the distribution of the pulp on the bottom-left part of the plot, contributing to the higher activity in both DPPH^•^ and ABTS^•+^ assays, while the radical scavenging activity of the albedo is related to the incidence of rutin, luteolin-*O*-rutinoside, and neoeriocitrin phenolic compounds, along with the limonoid isobacunoic acid. The rest of the phenolic compounds, limonoids, and the detected coumarins determined the discrimination of the peels on the right part of the plot. The incidence of compounds is highlighted in detail by the positive bars of the score contribution (Figure 2B).

## 4. Conclusions

A comprehensive chemical investigation on VOCs and non-volatile constituents of Faustrime was performed by GC-MS and UHPLC-QTOF-MS/HRMS analysis. Monoterpenes and sesquiterpenes were the most abundant VOCs identified, describing a citronellal/limonene/linalyl acetate chemotype. Coumarins (60–66), along with C-glycoside flavonoids (75, 78–79, 86), limonoids (67–70, 93, 95), and HMG-glycoside flavonoids (89, 91), were reported for the first time in the species. In particular, MS spectra of compound 91 was described for the first time in this species and tentatively attributed to isorhamnetin-di-HMG-*O*-glycoside. The metabolomic description of constituents of PE, PU, AL, and SE was defined by PCA plots highlighting the relevant occurrence of α-bisabolol, 2-menthen-1-ol, and α-*trans*-bergamotol VOCs in the cluster of pulp and seeds, whereas the non-volatile compounds were particularly relevant in the peel. The spectrophotometric assays for DPPH^•^ and ABTS^•+^ radical scavenging capacity showed the higher activity of the edible pulp and albedo, usually considered as a by-product of the fruit. The PCA modeling defined the incidence of citric acid in the discrimination of pulp and phenolic compounds rutin, neoeriocitrin and luteolin-*O*-rutinoside in the albedo. The antioxidant potential of the peels was attributed to the investigated coumarins and phenolic compounds. The consistent data obtained in the complex workflow of the foodomic approach defined the metabolome of bioactive compounds in the Faustrime fruits as a source of high-value constituents for future application in products with health-promoting effects.

## Figures and Tables

**Figure 1 antioxidants-12-00565-f001:**
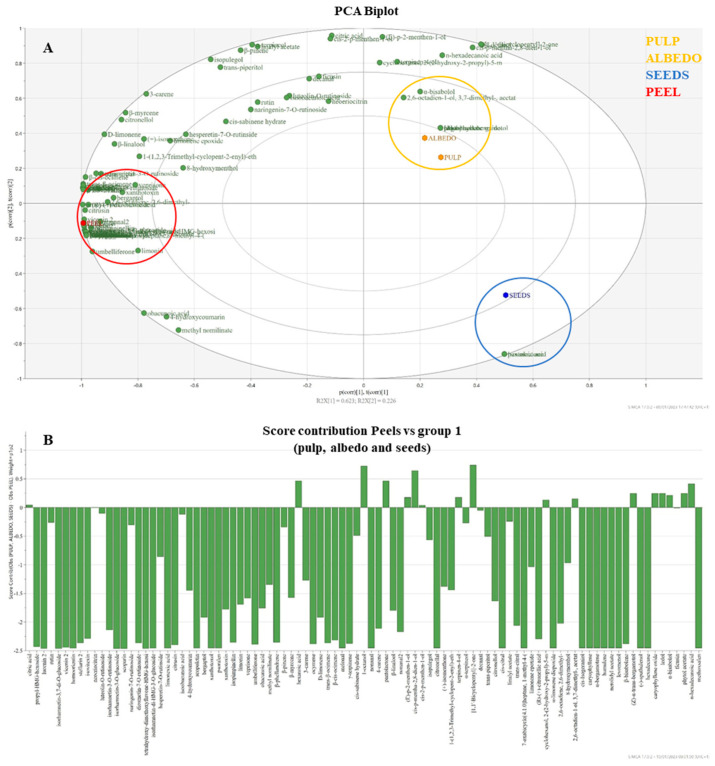
Biplot of PCA analysis performed on peels (red spot), pulp, albedo (yellow spot), and seeds (blue spot) for the identified VOCs and non-volatile compounds (**A**). Contribution diagram of the constituents on the distribution of samples (**B**).

**Figure 2 antioxidants-12-00565-f002:**
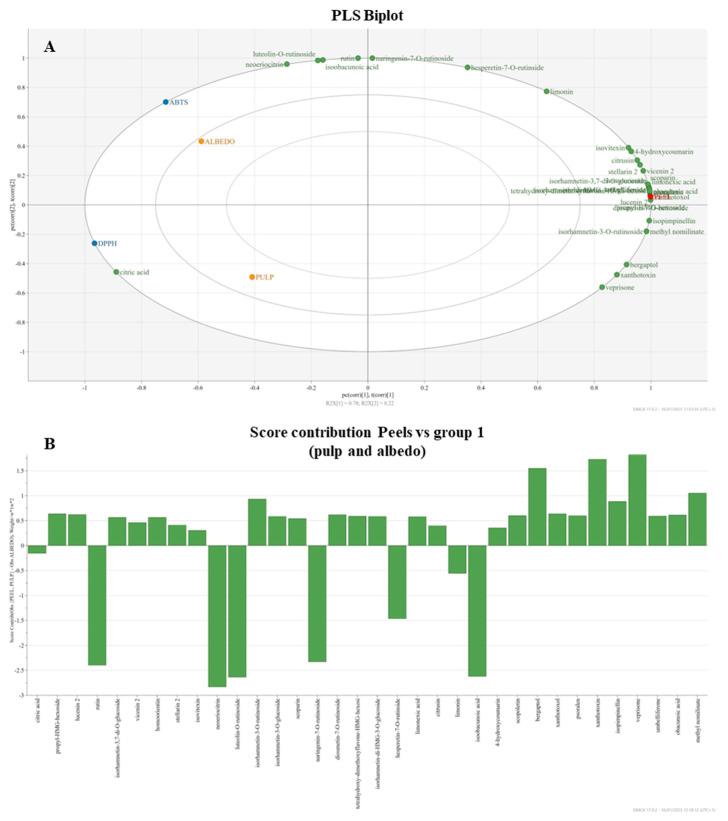
Biplot of PLS analysis performed on peels, pulp, and albedo for the identified non-volatile compounds (**A**). Contribution diagram of the constituents on the distribution of samples (**B**).

**Table 1 antioxidants-12-00565-t001:** GC-MS VOCs detected in peel, pulp, albedo, and seeds of Faustrime fruit.

			Relative Abundance (%)			
n. ^a^	t_R_	Compound	Peel	Seed	Pulp	Albedo
1	8.79	β-phellandrene	0.18	-	0.06	0.08
2	8.85	β-pinene	0.07	-	0.16	0.10
3	9.36	β-myrcene	0.33	-	0.40	0.25
4	9.49	hexanoic acid	-	9.59	-	-
5	9.85	3-carene	0.10	-	0.14	0.09
6	10.27	ocymene	2.40	-	0.94	0.92
7	10.40	limonene	12.99	8.08	11.40	9.03
8	10.70	*trans*-β-ocimene	0.32	-	0.14	0.12
9	10.98	β-cis-ocimene	0.09	-	0.06	0.03
10	11.12	melonal	0.06	-	-	-
11	11.25	γ-terpinene	3.73	-	1.75	1.34
12	11.49	*cis*-sabinene hydrate	0.08	-	-	0.14
13	11.67	1-octanol	-	-	0.17	0.12
14	11.70	nonanal	0.07	-	-	-
15	12.05	4-carene	0.14	-	-	0.08
16	12.23	pantolactone	-	6.70	-	-
17	12.43	β-linalool	3.11	-	3.73	1.42
18	12.44	nonanal	0.29	7.67	-	0.15
19	12.99	*trans*-p-2-menthen-1-ol	0.48	-	2.47	1.31
20	13.35	*cis*-p-mentha-2,8-dien-1-ol	-	-	0.28	0.31
21	13.51	*cis*-p-2-menthen-1-ol	0.46	-	1.67	0.88
22	13.62	isopulegol	0.31	-	0.47	0.46
23	13.88	citronellal	23.47	15.04	0.40	3.93
24	14.13	(+)-isomenthone	1.39	-	2.17	0.55
25	14.29	1-(1,2,3-trimethyl-cyclopent-2-enyl)-ethanone	0.26	-	-	0.27
26	14.47	terpinen-4-ol	0.72	-	5.07	1.70
27	14.85	α-terpineol	2.71	-	6.18	4.02
28	15.07	[1,1′-bicyclopentyl]-2-one	-	-	0.71	0.54
29	15.20	decanal	0.35	-	1.33	0.37
30	15.28	*trans*-piperitol	0.78	-	1.83	0.72
31	15.85	citronellol	10.65	7.67	8.60	9.68
32	16.10	*cis*-citral	1.15	-	-	-
33	16.45	linalyl acetate	7.65	14.62	18.23	10.60
34	16.83	*trans*-citral	1.74	-	1.68	0.46
35	17.15	7-Oxabicyclo[4.1.0]heptane, 1-methyl-4-(2-methyloxiranyl)	0.64	-	-	-
36	17.69	limonene epoxide	0.42	-	-	0.52
37	18.12	(R)-(+)-citronellic acid	2.62	-	-	1.02
38	18.44	cyclohexanol, 2-(2-hydroxy-2-propyl)-5-methyl-	4.46	19.82	-	-
39	18.62	α-limonene diepoxide	0.61	-	-	-
40	18.80	2,6-octadiene, 2,6-dimethyl-	0.91	-	0.81	-
41	18.96	8-hydroxymenthol	1.77	-	3.66	-
42	19.07	2,6-octadien-1-ol-3,7-dimethyl-acetate	0.57	10.81	-	9.33
43	19.44	*cis*-isogeraniol	0.26	-	-	-
44	20.39	caryophyllene	0.61	-	-	-
45	20.72	α-bergamotene	1.20	-	-	-
46	21.15	humulene	0.75	-	-	-
47	22.08	nerolidyl acetate	0.55	-	-	-
48	22.17	levomenol	0.27			
49	22.30	β-Bisabolene	1.75	-	0.59	
50	22.51	α-*trans*-bergamotol	-	-	0.45	-
51	22.81	(-)-spathulenol	0.83	-	-	-
52	24.15	hexadecane	0.16	-	-	-
53	24.49	caryophyllene oxide	-	-	0.66	-
54	25.37	ledol	-	-	1.14	-
55	25.92	α-bisabolol	0.23	-	2.96	0.53
56	28.52	ficusin	0.48	-	0.48	0.52
57	28.70	phytol acetate	-	-	0.43	-
58	30.71	*n*-hexadecanoic acid	0.43	-	4.96	8.36
59	31.70	methoxsalen	0.55	-	-	-

^a^ compounds are listed in ascending order of retention time.

**Table 4 antioxidants-12-00565-t004:** Semi-quantitative analysis of non-volatile compounds.

N	Compounds	Peel *	Pulp *	Albedo *
Chloroform Extract				
60	4-hydroxycoumarin ^a^	4.34 ± 0.25	1.78 ± 0.38	-
61	Scopoletin ^a^	2.34 ± 0.38	-	-
62	Bergaptol ^a^	9.63 ± 0.82	4.42 ± 0.25	-
63	Xanthotoxol ^a^	5.34 ± 0.41	1.90 ± 0.18	-
64	Psoralen ^a^	145.57 ± 9.65	6.95 ± 0.67	-
65	Xanthotoxin ^a^	35.00 ± 1.85	8.17 ± 0.68	-
66	Isopimpinellin ^a^	2.55 ± 0.14	1.54 ± 0.21	-
Hydroalcoholic Extract				
75	lucenin 2 ^b^	10.78 ± 1.37	6.91 ± 0.38	6.92 ± 0.43
76	isorhamnetin-3,7-di-*O*-glucoside ^c^	60.60 ± 4.62	9.77 ± 0.75	13.60 ± 0.87
77	vicenin 2 ^d^	19.38 ± 2.15	3.94 ± 0.11	3.46 ± 0.15
78	homoorientin ^b^	4.50 ± 0.98	1.83 ± 0.20	2.28 ± 0.29
79	stellarin 2 ^c^	27.56 ± 2.05	12.06 ± 0.65	25.60 ± 1.08
80	rutin ^e^	94.45 ± 5.13	52.17 ± 3.44	34.50 ± 2.12
81	isovitexin ^d^	2.30 ± 0.13	1.36 ± 0.09	1.67 ± 0.14
82	neoeriocitrin ^f^	9.32 ± 0.79	4.80 ± 0.32	10.92 ± 0.82
83	luteolin-*O*-rutinoside ^b^	11.62 ± 0.99	6.77 ± 0.59	7.95 ± 0.71
84	isorhamnetin-3-*O*-rutinoside ^c^	12.96 ± 0.81	10.55 ± 0.83	8.74 ± 0.52
85	isorhamnetin-3-*O*-glucoside ^c^	11.32 ± 0.98	8.31 ± 0.79	10.25 ± 1.05
86	scoparin ^c^	13.56 ± 0.85	9.67 ± 0.67	8.58 ± 0.96
87	naringenin-7-*O*-rutinoside ^g^	25.86 ± 1.54	2.74 ± 0.13	3.63 ± 0.36
88	diosmetin-7-*O*-rutinoside ^c^	7.34 ± 0.71	11.33 ± 1.03	7.41 ± 0.81
89	tetrahydroxy-dimethoxyflavone HMG-hexoside ^c^	10.77 ± 0.89	5.11 ± 0.38	7.61 ± 0.77
90	hesperetin-7-*O*-rutinside ^h^	19.87 ± 1.11	5.07 ± 0.42	15.55 ± 1.28
91	isorhamnetin-di-HMG-*O*-glycoside ^c^	38.18 ± 2.98	11.33 ± 0.95	6.75 ± 0.56

Calculated using the calibration curves of: ^a^ coumarin; ^b^ luteolin; ^c^ quercetin; ^d^ apigenin; ^e^ rutin; ^f^ eriocitrin; ^g^ naringenin, and ^h^ hesperetin. * data are expressed as mg std equivalent/g of dried extract (EXT).

**Table 5 antioxidants-12-00565-t005:** DPPH^•^ and ABTS^•+^ radical scavenging capacity of hydroalcoholic and chloroform extracts.

Part	Extract	DPPH^•^	ABTS^•+^
		mean TE ± SD	mean TE ± SD
Peel	Hydroalcoholic	2.75 ± 0.05	8.22 ± 0.32
	Chloroform	2.06 ± 0.04	5.01 ± 0.11
Pulp	Hydroalcoholic	7.58 ± 0.12	10.51 ± 0.34
	Chloroform	1.73 ± 0.02	3.61 ± 0.11
Albedo	Hydroalcoholic	7.71 ± 0.32	10.82 ± 0.45
	Chloroform	1.15 ± 0.11	7.73 ± 0.26
Seeds	Hydroalcoholic	3.64 ± 0.23	4.19 ± 0.12
	Chloroform	3.07 ± 0.15	1.21 ± 0.09

mean TE values are expressed as µg of standard Trolox solution exerting the same radical scavenging activity of 1 mg/mL of the tested extract. SD is the standard deviation of three independent experiments.

## Data Availability

The original contributions presented in this study are included in the article. Further inquiries can be directed to the corresponding author.

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
