# Peer review of "LC-MS and GC-MS Data Fusion Metabolomics Profiling Coupled with Multivariate Analysis for the Discrimination of Different Parts of Faustrime Fruit and Evaluation of Their Antioxidant Activity"

_antioxidants, 2023, doi:10.3390/antiox12030565_

Round 1

Reviewer 1 Report

Observations regarding the article „LC-MS and GC-MS data fusion metabolomics profiling coupled with multivariate analysis for the discrimination of different parts of Faustrime fruit and evaluation of their antioxidant activity” authors: Ciro Cannavacciuolo et al.,

The authors have conducted a highly scientific study that provides information on the presence of bioactive compounds in Faustrime fruits as a source of constituents for their application in products with health-promoting effects. Also, the data obtained were analysed using PCA and PLS models, showing correlations between the identified compounds. But, some other observations can be made, namely

  1. The meaning of the following terms should be mentioned in the Abstract: VOCs, PCA, PLS....
  2. The extraction of volatile and non-volatile compounds was carried out using certain extraction times, 2 days and 3 days respectively. Are there standard extraction times for these compounds?

Author Response

Thank you for your observation. The maceration extraction using solvents at increasing polarity is performed usually for 3 days as widely reported in literature (Cannavacciuolo et al., 2022 (https://doi.org/10.1016/j.foodchem.2022.132968). Regarding the extraction of VOCs with Hexane to reduce toxic solvent and avoid possible degradation of compounds  we preferred to follow the optimized protocol reported by Guzowska et al., 2022 (doi.org/10.3390/molecules27092911). In this paper the authors point out that there is no difference between 2 and 3 days, therefore we preferred used this extraction condition.

Reviewer 2 Report

This is a very interesting work describing LC-MS and GC-MS data fusion metabolomics profiling in combination with chemometrics to discriminate and characterize different parts of Faustrime fruit. The manuscript is, in general, well written, the experiments correctly performed, and the results well discussed. The research is interesting for the scientific community working in this field and in my opinion, the manuscript is suitable for publication in Antioxidants after a revision.

Lines 13-14: Please, indicate also the fruit… not only the parts you are studying…

Line 44: Change “then” by “the”

Line 69-71: The better way of describing the HRMS methodology will be “UHPLC-Q-TOF-MS/HRMS”. When tandem mass spectrometry is performed using a Q-TOF instrument, precursor ion is isolated in low resolution (MS) and then product ions are analyzed in high resolution (HRMS), for this reason, normally, the tandem mass spectrometry experiment must be noted as MS/HRMS and not MS/MS.

Line 106: Use “µm” instead of “um”

Lines 106-108: Please, check consistency with the units, sometimes they are separated from the number, others not (this is the case with Celsius degrees in this sentence). Check through all the manuscript and all the units, and try to be consistent.

Line 131-132: Explain better what do you mean by the sample concentration was 0.5 mg/L. I suppose that you mean 0.5 mg of sample extract/L. This needs to be better explained….

Line 133: Change “adopted” by “used”, “employed”…

Line 135: HRMS/MS is not correct, although commonly people employ this notation when they do tandem mass spectrometry with HRMS instruments, in fact, as previously commented, when a Q-tof instrument is employed, precursor ion is isolated in the quadrupole (low-resolution analyzer), and the product ions analyzed in the TOF (high-resolution analyzer), so the correct notation to describe the tandem mass spectrometry experiment is MS/HRMS. Be consistent with this comment throughout the manuscript.

Lines 141-142: Same comment as before, you are not doing MS/MS experiments (this means that both analyzers are of low-resolution characteristics).

Lines 189-189: You can also comment that the use of a QC will also guarantee the robustness of the chemometric results. If quality controls are not grouped, chemometric results cannot be trusted.

Line 207: Please, correct 3.1 instead of c3.1

Line 210: Please, describe how this relative abundance was really calculated. In any case, you also must comment that you are supposing that the response factor is the same for all the detected chemicals, which in fact is not correct. At least, this point needs to be addressed or commented on in the text…  A higher relative abundance value does not directly mean a higher content if you are comparing compounds with different response factors. And this is especially important when MS instruments are employed. According to that, when discussing compound levels, please do not use terms such as… this compound is the most abundant… because it may be wrong if you have not quantified the real concentration… Better that you say … this compound shows the higher relative abundance… (if you previously have explained how this relative abundance was calculated). Be consistent with this comment throughout the manuscript.

Line 255: Please, do not use MS2 term. This terminology is normally employed when tandem mass spectrometry is performed with ion-trap instruments, so the tandem mass experiments are carried out in time, not in space such as in the case of q-tof instruments. In that case, what you have done is MS/HRMS. So you can say MS/HRMS spectra.

-          Same comment for Tables 2 and 3. Is not MS/MS, it is MS/HRMS

Section 3.3. This is not a quantitative analysis if you are not quantifying each compound with its corresponding standard. What you are doing is a semi-quantitative analysis based on quantification using a different standard, which is not wrong, and typically done in this kind of study (as normally we do not have access to many pure chemical standards), but the employed terminology must be correct. So, please, do not say quantitative analysis, and refer always to a semi-quantitative analysis. And to be accurate, you also need to indicate which are the 8 standards employed, and which one was employed for the semiquantitative analysis of each compound.

Supplementary information must also be indicated at the end of the manuscript. This section is missing.

Author Response

Reviewer 2

This is a very interesting work describing LC-MS and GC-MS data fusion metabolomics profiling in combination with chemometrics to discriminate and characterize different parts of Faustrime fruit. The manuscript is, in general, well written, the experiments correctly performed, and the results well discussed. The research is interesting for the scientific community working in this field and in my opinion, the manuscript is suitable for publication in Antioxidants after a revision.

Lines 13-14: Please, indicate also the fruit… not only the parts you are studying…

Thank you for your observation, the sentences has been modified according to your suggestion.

Line 44: Change “then” by “the”

The sentence was corrected.

Line 69-71: The better way of describing the HRMS methodology will be “UHPLC-Q-TOF-MS/HRMS”. When tandem mass spectrometry is performed using a Q-TOF instrument, precursor ion is isolated in low resolution (MS) and then product ions are analyzed in high resolution (HRMS), for this reason, normally, the tandem mass spectrometry experiment must be noted as MS/HRMS and not MS/MS.

Thank you for the bright consideration for improving mass spectrometry technical terms. The acronym MS/HRMS was substituted according to your correction.

Line 106: Use “µm” instead of “um”

Sorry the typing mistake, however the sentence has been corrected.

Lines 106-108: Please, check consistency with the units, sometimes they are separated from the number, others not (this is the case with Celsius degrees in this sentence). Check through all the manuscript and all the units, and try to be consistent.

According also with editors comment which requires us to express the units separated, we have used this layout to express all the units in whole manuscript.

Line 131-132: Explain better what do you mean by the sample concentration was 0.5 mg/L. I suppose that you mean 0.5 mg of sample extract/L. This needs to be better explained….

Actually, the “sample concentration was 0.5 mg/L” could be misunderstood, the mean of the concentration was referred to 0.5 mg of extract in one mL of solvent. Therefore, to avoid misunderstandings the meaning was clarified in the text.

Line 133: Change “adopted” by “used”, “employed” …

The change has been modified in the manuscript.

Line 135: HRMS/MS is not correct, although commonly people employ this notation when they do tandem mass spectrometry with HRMS instruments, in fact, as previously commented, when a Q-tof instrument is employed, precursor ion is isolated in the quadrupole (low-resolution analyzer), and the product ions analyzed in the TOF (high-resolution analyzer), so the correct notation to describe the tandem mass spectrometry experiment is MS/HRMS. Be consistent with this comment throughout the manuscript. Lines 141-142: Same comment as before, you are not doing MS/MS experiments (this means that both analyzers are of low-resolution characteristics).

Thank you for your consideration. The mass spectrometry technical term MS/HRMS was corrected along all the manuscript long.

Lines 189-189: You can also comment that the use of a QC will also guarantee the robustness of the chemometric results. If quality controls are not grouped, chemometric results cannot be trusted.

Thank you for the suggestion, However the description of the use of QC and its its ability to give robustness at chemometric results have been reported in the manuscript.

Line 207: Please, correct 3.1 instead of c3.1

The typing mistake has been corrected according to reviewer suggestion was corrected.

Line 210: Please, describe how this relative abundance was really calculated.

For a GC MS analysis the relative abundance (%) is calculated with following equation dividing the area of each individual peak by the total sum of all detected peak area multiplied by 100. However, this equation is automatically calculated by the GC instrument software and generate in the report processing file, regarding the sum of each column that is not 100% this results its generate because we removed the unknow compounds from the table.

In any case, you also must comment that you are supposing that the response factor is the same for all the detected chemicals, which in fact is not correct. At least, this point needs to be addressed or commented on in the text…  A higher relative abundance value does not directly mean a higher content if you are comparing compounds with different response factors. And this is especially important when MS instruments are employed. According to that, when discussing compound levels, please do not use terms such as… this compound is the most abundant… because it may be wrong if you have not quantified the real concentration… Better that you say … this compound shows the higher relative abundance… (if you previously have explained how this relative abundance was calculated). Be consistent with this comment throughout the manuscript.

The results of semiquantitative analysis in whole manuscript have been expressed as rightly suggested by the reviewer

Line 255: Please, do not use MS2 term. This terminology is normally employed when tandem mass spectrometry is performed with ion-trap instruments, so the tandem mass experiments are carried out in time, not in space such as in the case of q-tof instruments. In that case, what you have done is MS/HRMS. So you can say MS/HRMS spectra.

-          Same comment for Tables 2 and 3. Is not MS/MS, it is MS/HRMS

Thank you for your consideration. The mass spectrometry technical term MS/HRMS was corrected all the manuscript long.

Section 3.3. This is not a quantitative analysis if you are not quantifying each compound with its corresponding standard. What you are doing is a semi-quantitative analysis based on quantification using a different standard, which is not wrong, and typically done in this kind of study (as normally we do not have access to many pure chemical standards), but the employed terminology must be correct. So, please, do not say quantitative analysis, and refer always to a semi-quantitative analysis. And to be accurate, you also need to indicate which are the 8 standards employed, and which one was employed for the semiquantitative analysis of each compound.

The terminology was changed to semi-quantitative according to your suggestions in the manuscript. Moreover, a legend of the standard employed for the quantitative analysis was added at the bottom of table 4.

Supplementary information must also be indicated at the end of the manuscript. This section is missing.

The information regarding the downloading of supplementary material has been insert in the manuscript

Round 2

Reviewer 2 Report

The authors correctly addressed all my comments and suggestions. The manuscript can be accepted in its present form.